# Bioactivity and Therapeutic Potential of Kaempferol and Quercetin: New Insights for Plant and Human Health

**DOI:** 10.3390/plants11192623

**Published:** 2022-10-05

**Authors:** Rahmatullah Jan, Murtaza Khan, Sajjad Asaf, Saleem Asif, Kyung-Min Kim

**Affiliations:** 1Department of Applied Biosciences, Graduate School, Kyungpook National University, Daegu 41566, Korea; 2Coastal Agriculture Research Institute, Kyungpook National University, Daegu 41566, Korea; 3Department of Horticulture and Life Science, Yeungnam University, Gyeongsan 38541, Korea; 4Natural and Medical Sciences Research Center, University of Nizwa, Nizwa 616, Oman; 5Department of Botany, Abdul Wali Khan University Mardan, Mardan 23200, Pakistan

**Keywords:** kaempferol, quercetin, therapeutic, bioavailability, bioactivity, antioxidant

## Abstract

Plant secondary metabolites, especially flavonoids, are major metabolites widely found in plants that play several key roles in plant defence and signalling in response to stress conditions. The most studied among these flavonoids are kaempferol and quercetin due to their anti-oxidative potential and their key roles in the defence system, making them more critical for plant adaptation in stress environments. Kaempferol and quercetin in plants have great therapeutic potential for human health. Despite being well-studied, some of their functional aspects regarding plants and human health need further evaluation. This review summarizes the emerging potential of kaempferol and quercetin in terms of antimicrobial activity, bioavailability and bioactivity in the human body as well as in the regulation of plant defence in response to stresses and as a signalling molecule in terms of hormonal modulation under stress conditions. We also evaluated the safe use of both metabolites in the pharmaceutical industry.

## 1. Introduction

Flavonoids are secondary metabolites found ubiquitously throughout plants. Flavonoids contain a large group of compounds that can be divided in to many major subgroups such as, chalcones, flavones, aurones, isoflavonoids, flavonols, flavandiols, flavones, anthocyanins, proanthocyanidins or condensed tannins. More details of classification were enumerated in a previously published review [1,2,3]. The flavonoids contain several structural classes, but they have the same 3-hydroxyflavone skeleton (Figure 1A). Each class of flavonoids is defined based on the substitution of the C ring while the substitution on A and B ring defines each member of individual class [4]. In this way, the phenolic substitution at C-2 position and hydroxyl substitution at C-3 position constitute kaempferol and quercetin respectively (Figure 1B,C). In humans, these compounds are related to a large range of health benefits arising from their bioactive properties, such as anti-inflammatory, anticancer, anti-aging, cardio-protective, neuroprotective, immunomodulatory, antidiabetic, antibacterial, antiparasitic, and antiviral properties [5,6,7]. In plants, flavonoids are involved in a wide range of activities such as, cell growth regulation, protecting plants against biotic and abiotic environmental factors, and attracting pollinators [8]. On the basis of chemical structure, flavonoids are divided into six classes such as, anthocyanidins, flavanols, flavanones, flavones, flavonols, and isoflavones [9]. Among the flavonoids, flavonols are the most frequent flavonoid chemical structures, responsible for giving food colour and flavour, preventing the oxidation of fat, and protecting vitamins and enzymes [10]. Kaempferol and quercetin are the main class of flavonols and differ from each other by OH group; that is, quercetin has an extra OH group at the third position of the B ring. They are abundantly present in onions, apple and broccoli [11,12]. The structural variation in most of the flavonoids are due to the changeover of the differentially located hydrogen ion with other group such as, hydroxyl, methoxyl, and glycosyl. Attachment of galactose at the same portion generates another derivative, named quercetin *3*-*O* galactoside or hyperoside. Likewise, the rhamnosyl group addition to the 3-OH or 7-OH group results in the development of quercetin *3*-*O* rhamnoside and quercetin *7*-*O*-rhamnoside, respectively. Disaccharides like glucose and rhamnose are also attached to quercetin and form another derivative known as rutinose or *α-L*-rhamnopyranosyl-(*1 → 6*)-*β*-*D*-glucopyranose. Rutin is also a vital derivative, having disaccharides at the 3-OH position. The information of quercetin derivatives are published in more details in the review of Priyanka et al., (2021) [13].

Kaempferol and quercetin bioactivity in human body is dependent on the bioavailability to reach to the body tissues. Bioavailability of a compound refers to the digestion level, absorption and metabolism after the food ingestion, which is a potential step to evaluate the mechanism of action of a compound [14]. Some pharmaceutical studies of kaempferol and quercetin show that they are more efficiently bioavailable in conjugate form rather than the free form [15,16,17]. Recent literature concludes that the absorption of kaempferol and quercetin quickly metabolize in the liver. From the liver it circulates in the form of methyl, glucuronide and sulfate conjugate in the whole body [18]. The bioactivity of both these flavone can be assessed by measuring these conjugates in the blood and urine of human.

Flavonoids act as strong antioxidants in plants challenged by various environmental stresses. It has been understood for decades that the increase in flavonoid metabolism protects plants against environmental constraints. The generation of reactive oxygen species (ROS) in response to abiotic stress is a common and rapid action and can be a converging point for stress signalling and defence responses [19]. Commonly, plants increase flavonoid biosynthesis in response to the stress-induced generation of ROS [20,21]. There are few points of argument regarding stress reduction by flavonoids. For instance, flavonoids are found in the vacuole and are translocated from the site of biosynthesis to the site of ROS production. Additionally, flavonoids are also found mostly in epidermal tissue and thus cannot protect most of the plant tissues significantly. Furthermore, plant cells have efficient defence system that successfully inhibit ROS generation and therefore the antioxidant role of flavonoid would be redundant [22]. However, recent studies on model plant other plant species have unravelled that the in vivo function of flavonoid antioxidant is important for the survival of plants under stress conditions [22,23]. Additionally, recent studies suggested that based on their structure, flavonoids are the strongest antioxidants induced by most environmental stresses [23].

Kaempferol and quercetin are the two main flavonoid species abundantly present in plants. As a result, they are significantly gaining attention in the plant research associated with abiotic stress because of their antioxidant effects. In the current review, we evaluated the recent advances regarding kaempferol and quercetin as plant stress inhibitors.

## 2. Kaempferol and Quercetin Contents in Edible Plants

Flavonoids are diphenylpropanes found most commonly in edible plants which are a common component in the human diet. Kaempferol and quercetin are the main flavonoids, and despite being medicinally important, they have been given a little attention to in their constitution in the human diet. It is estimated that about 1 g/day total flavonoids are to be consumed for a proper diet [24]. Although, no dietary recommendation of flavonol intake has been established for individuals, in the US, it is reported to range from 9.0 to 36.2 mg/day, which is lower than the UK and other European countries [25,26,27]. In the US, tea, onion, apple and red wine contribute more flavonol to the human diet while average kaempferol and quercetin uptake is about 3.5 and 5.4 mg/day, respectively [11,28]. Generally, the level of processed flavonoids is reduced by 50% compared to fresh foods, therefore fresh foods should be prepared [29].

Flavonoids, especially kaempferol and quercetin are among the bioactive compounds that contribute to human health. As it is reported, kaempferol and quercetin has cardioprotective and antihypertensive benefits in human [30]. Mostly, vegetables and few other plants are the rich source of kaempferol and quercetin as presented in Table 1, reported by Koo Hui Miean and Suhaila Mohamed [31]. They detected high concentration of quercetin in a few plants such as, *Camella chinensis* (1070 mg/kg), *Allium fistulosum* (1497 mg/kg), *Carica papaya* (811 mg/kg), *Capsicum annum* (799 mg/kg), and kaempferol in *Allium fistulosum* (832 mg/kg), *Carica papaya* (453 mg/kg), and *Cucarbita maxima* (371 mg/kg). It is shown in the Table 1 that plants synthesize high content of quercetin than the kaempferol. The lowest level of quercetin was found in *Brassica alboglabra*, *Raphanus sativa*, *Sesbania grandifolia* and *Phaeomeria speciose* (14, 17, 18 and 21 mg/kg respectively), while the lowest level of kaempferol were found in *Hydrocotyl asiatica* (20 mg/kg) and *Sesbania grandifolia* (21 mg/kg) [31]. Another study evaluated that onion, asparagus, and berries contain high concentration of quercetin, while leeks, apple, and chives contain low level of quercetin [18]. On the other hand, spinach and kale are the richest source of kaempferol, and apple, cranberry, and lettuce contain the lowest levels of kaempferol [18]. The leaves of wild leeks were investigated as containing 50 and 32 mg/100 g of fresh weight of quercetin and kaempferol respectively [32]. The concentrations of any kind of flavonoid vary between species to species and even from an environment to another environment. The concentrations of flavonoids depend on several factors such as growing condition, growing environment, stage of development, and parts of plant. Reports show that fruit and vegetables are the richest sources of kaempferol and quercetin, and therefore, the consumption of the fruit and vegetables in the diet significantly increases the antioxidant capacity [33]. It is recommended to utilize more vegetables and fruits in the diet instead of taking antioxidants as a supplement to protecting the body against oxidative stress. Several studies have established the natural antioxidants flavonoids found in edible plants such as allium, tomato, lettuce, cauliflower, radish, pea, broccoli, Chinese cabbage, carrot, colocasia, kale, chive, garlic, orange, grapefruit, lemon, apple, pineapple, peach, apricot, pear, and grape [31]. In short, fruit and vegetable are rich source of kaempferol and quercetin and are recommended to take as natural antioxidant supplements.

Furthermore, glycosylation of flavonoid compounds increases their solubility in water, which results into increase bioavailability to improve their pharmaceutical properties [34]. Solubility in aqueous and organic solvent is an important property of flavonoid. Flavonoids are poorly soluble in aqueous solution and can easily oxidize their forming insoluble polymer [35]. Due to their phenolic group, flavonoids are weak acidic, and they are easily soluble in alkaline solution; however their dietary sources are mostly acidic. While the solubility of flavonoids in lipophilic solvents partially prevents them from oxidation in food sources, their solubility in water generally improves their bioavailability from the diet and glycosylation usually increases aqueous solubility [36]. The presence of sugar moieties usually leads to the enhanced bioavailability of the respective flavonoid aglycone depending on the nature of the sugar. For example the sugar containing flavonol glucosides like kaempferol 3-*O*-glucoside (astragalin) and quercetin 3-*O*-glucoside (isoquercetin) get absorbed quickly than the other glucosides in the human gut because hydrolyzing enzymes for sugar glucosides are present in the human intestine [37]. Studies stated that, quercetin from isoquercetine has more absorbability than rutin and quercitrin in rats, pigs and spiraeoside in human [34]. However, the glycosylated flavonoid reduces the antioxidant efficiency compared to the corresponding aglycones due to increasing the number of glycosidic moieties and position of sugar [34]. A report shows the glycosylation not only reduce antioxidant activity but can also inhibits anti-inflammatory, antibacterial, antifungal, anticoagulant, and anti-tubercular activity, while promoting certain biological activity such as anti-rotavirus, anti-obesity, anti-adipogenic and anti-allergic activity, and anticholinesterase potential [38].

## 3. General Mechanism of Stress Inhibition by Kaempferol and Quercetin

Flavonoids have a key importance in plants and play an important role in physiological process including, defence against pest, pathogens and many other environmental stresses. Plant activate antioxidant system in response to excessive generation of reactive oxygen species defence [39,40]. Flavonoids are also well studied against plant biotic and abiotic stresses. However, among them kaempferol and quercetin are not well studied, and little is known of the mechanism of kaempferol/quercetin-mediated immunity. Kaempferol and quercetin glycosylated into glycosidic derivatives which can scavenge free radicles by donating an electron or hydrogen [41,42,43]. The glycosides of kaempferol and quercetin, including kaempferol-3,7-dirhamnoside (KRR), kaempferol 3-*O* glucoside 7-Orhamnoside (KGR), kaempferol 3-*O*-[6″-*O*-(rhamnosyl) glucoside] 7-*O* rhamnoside (KRGR), quercetin 3-*O*-glucoside 7-*O*-rhamnoside (QGR), quercetin 3-*O*-[6″-*O*-(rhamnosyl) glucoside] 7-*O*-rhamnoside (QRGR), and quercetin 3-*O*-rhamnoside 7-*O*-rhamnoside (QRR), are either involved in the direct defence mechanism or indirect defence mechanism [44]. Studies have revealed that, both the kaempferol and quercetin have great antioxidant activity and possess efficient ROS scavenging property. Kr contains a 3-OH group, while quercetin contains 3 and 5-OH groups in the C and A rings, respectively, with the capacity to donate the most electrons [45,46]. Kaempferol, quercetin and apigenin were found to be involved in auxin-regulated cell division as a signalling molecule [47,48]. Quercetin generated in the plant epidermal tissue to protect the tissue from oxidative stress are induced by high light intensity [49]. In the previous reports, it is evaluated that environmental stress induces flavonone 3-hydroxylase (*F3H)* gene which enhances the accumulation of kaempferol, quercetin and anthocyanin, which alter an interaction between jasmonic acid (JA), gibberellic acid (GA), slender (*SLR)* and DELLA protein to cope with stress condition [50]. We recently reported that pathogenic stress induces *F3H* gene which enhance the accumulation of kaempferol and quercetin [51]. The exogenous application of kaempferol and quercetin act as significant pesticides against white-backed planthopper while, their glycosides act as strong inhibitors of *Nilaparvatalugens* digestion, acting as deterrent [51,52]. Our previous report shows that, accumulation of kaempferol glycosides, induces activation of *SLR1* gene mediated via anthocyanin which suppress GA biosynthesis via activation of DELLA protein [50]. It is investigated that *SLR1* significantly induces salicylic acid (SA) and JA which results into regulation of defence system [53]. SA is one of the key plant hormone that reduces plant stress and enhance plant growth as well as the reduction in lipid peroxidation and superoxide contents. SA accumulation reduces oxidative injuries via increasing ROS scavenging antioxidant enzymes and also enhances chlorophyll contents [54]. While overaccumulation of quercetin significantly increases SA regulation and reduces JA accumulation which indicates that quercetin promotes SA and JA antagonistic association [55]. SA and JA has a key role in plant defence system where its activity depends on the invading pathogen. SA plays a major part in basal resistance to bacterial infection, whereas JA responds to fungal pathogens [54]. In short, quercetin can also regulate plant defence system via SA induction. Quercetin mediated induction of SA induces PR genes which also enhances stress tolerance. In the previous study it was assumed that, kaempferol and quercetin glycosides were toxic to pathogen or it promote programmed cell death and stomata closure mediated by SA accumulation [50]. In some stress condition, plants convert kaempferol and quercetin glycosides into anthocyanin, which promote *SLR1* gene expression and suppress DELLA protein and GA biosynthesis. Anthocyanin also inhibit the JA pathway due to the inhibition of Col-1unit of conjugates of Col-1+ JA Ile, which regulates the expression of JA responsive genes [50]. The possible ways of kaempferol and quercetin involvement in plant defence system is significantly drown in the proposed model (Figure 2). Briefly, the environmental constrains activate the flavonoids biosynthesis pathway by induction of key genes such as *F3H*, *FLS*, *DFR* and *SLR*. The transcriptional regulation of flavonoids biosynthesis related genes enhances the biosynthesis of kaempferol and quercetin that results into either scavenging the ROS or regulate SA biosynthesis. The SA accumulation also reduces ROS generation, enhances programed cell death and induces PR genes. On the other hand, the glycosides of kaempferol and quercetin either act as a toxic to the pathogen or inhibit the digestibility of the tissue. The DFR gene induced in the stress condition enhances the biosynthesis of anthocyanin, which play a key role in growth and development under stress condition.

## 4. Flavonoid Role as Growth Regulators in Plant

Phytohormones during stress condition coordinate with the biosynthesis of flavonoids like kaempferol and quercetin as shown in the scheme in Figure 3 [56]. Kaempferol and quercetin are not involved in growth regulation directly but they act as indirect growth regulators. Abscisic acid (ABA), salicylic acid (SA), jasmonic acid (JA), ethylene (ET), and auxin are the major phytohormones involved in plant growth under stress condition. It is reported that flavonoids (kaempferol and quercetin) regulate ROS signalling which in turn modulates hormonal (auxin, ABA) signalling [57,58]. An In vitro study showed that kaempferol and quercetin compete with naphthylphthalamic acid (a synthetic auxin transporter inhibitor) and can disturb auxin transport in various tissues of plant [59]. The role of flavonoids as auxin transport inhibitor was also investigated in arabidopsis in in vivo study [60]. In vivo experiments on Arabidopsis thaliana shoots showed that a specific flavonol bis-glycoside (i.e., kaempferol *3*-*O*-rhamnoside-7-*O*-rhamnoside) acted as an endogenous polar auxin transport inhibitor thus reducing plant stature [61]. It is shown in the Figure 3 that, quercetin reduces the auxin transport capacity of ATP-binding cassette type B (ABCB) families and PIN-FORMED (PIN) (auxin efflux transporters) [57,58]. In some cases, kaempferol and quercetin can also change the ROS level which can regulate polar auxin transport [62]. Kaempferol and quercetin has a key role in abiotic stresses such as salt, drought and heat stress as they regulate auxin distribution by regulating auxin flows and hence contribute in the control of plant development under stress condition. It is investigated that accumulation of auxin promote development of root architecture [63].

ABA is another phytohormone with a key role in physiological processes and promote plant adaptation to drought, salt and other environmental stresses. Kaempferol and quercetin are also inter-related to ABA biosynthesis. It was investigated in a report that flavonols regulate ABA signalling and ABA regulate flavonols biosynthesis [64]. In plants, *R2R3-MYB* is a gene family associated with redox control which can be expressed by regulation of ABA signalling [65,66]. MYB is major transcriptional factor family associated with the plants flavonoids biosynthesis, which evaluate the connection between ABA signalling and flavonoid biosynthesis [67]. Recently, it is found that quercetin modulate ABA signalling cascade. It is a fact that ABA induces stomata closure mediated via ROS generation in guard cells while, kaempferol and quercetin scavenge the ROS and reduce the risk of cell damage and enhance stomata closure in response to ABA [68]. This complex interaction of ABA mediated ROS generation and scavenging mechanism of ROS by kaempferol and quercetin in response to stress condition thus needs to be studied further and understood.

**Figure 3 plants-11-02623-f003:**
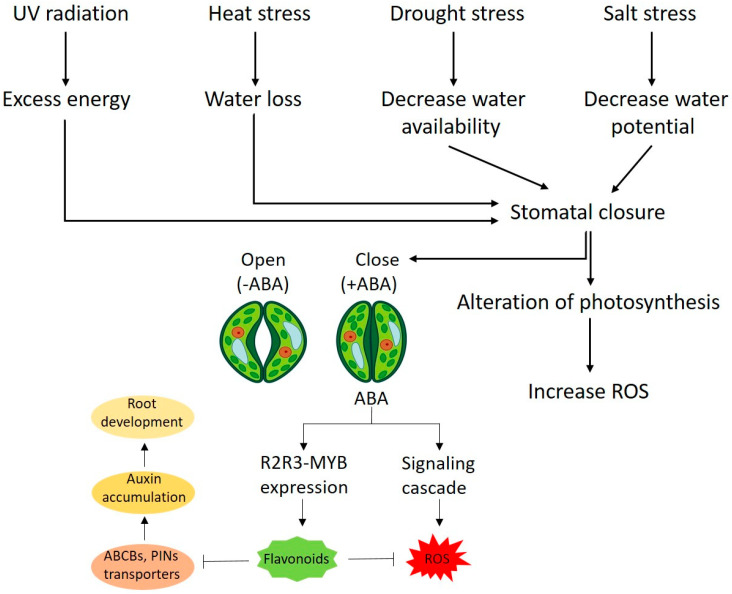
Proposed model indicate schematic representation of plant responses to abiotic stresses mediated via ABA and flavonoid signalling. Mostly, abiotic stress reduces water availability and consequently plant close the stomata and release ABA hormone. Flavonoids (kaempferol, quercetin) act as ROS scavenger and interact with ABA and auxin in leaf and root respectively. Stomatal opening in leaves is permitted by ABA binding to the membrane receptors resulting in ions and water efflux leading to stomata closure. ABA induces *R2R3-MYB* gene and stimulating flavonoids biosynthesis, which also triggers a signalling cascade which leading to ROS generation. In root, flavonoids reduce auxin transport resulting to auxin accumulation and root development. This model is adopted from Justine Laoué et al. [69].

## 5. Anti-Bacterial Activity of Kaempferol and Quercetin

Kaempferol and quercetin are found in a wide range of plant species. In recent decades, they have grown in importance due to their profitable bioactive benefits such as, anti-viral, anti-bacterial, anti-fungal, anti-inflammatory and cardioprotective properties [70]. Recently, both has been proved by United States Food and Drug Administration (USFDA) as a safe (GRAS) drug [71]. The mechanism of antimicrobial action of these phytochemicals has been widely explored in order to use them effectively in drug development. Pharmaceutical study shows that kaempferol and quercetin are potential antimicrobial agents and inhibits various pathogenic microorganisms. Microbial studies provide interesting literatures on the mechanism of kaempferol and quercetin. It was shown that kaempferol acts to destroy the activity of pathogen (*Staphylococcus aureus*) and hinders the anchoring of surface protein which reduce the adhesion of fibrinogen that promotes biofilm formation [72]. Another possible mechanism of pathogen growth inhibition is related to inhibition of gene expression involved in biofilm formation. Kaempferol reduces the gene expression level related to biofilm formation such as, clump factor A and B, (*clfA* and *clfB*), fibronectin-binding proteins A and B (*fnbA*, *fnbB*), and staphylococcal accessory regulator A (*sarA*), in *Staphylococcus aureus* [72]. In some pathogens like *S. pyogenes*, *S. aureus*, *S. haemolyticus*, *S. pyogenes*, *E. coli*, and *K. pneumonia*, quercetin increases inhibitory influences by increasing its cytoplasmic membrane permeability [73,74]. Hossion et al., [75] investigated that quercetin glycoside act as a novel antibacterial agent and inhibits DNA gyrase and topoisomerase IV. Report shows that kaempferol and quercetin inhibit the interaction of DNA binding helicase and deoxynucleotide triphosphates which results into *K. pneumonia* growth inhibition [76]. Further Huang et al. [77] elaborated that kaempferol inhibits the DNA PriA helicase of *S. aureus*, and the concentration of phosphate from ATP hydrolysis by this DNA helicase was decreased significantly in the presence of kaempferol. These reports suggests that, kaempferol and quercetin can bind to DNA helicase and inhibit its ATPase activity, which is a new mechanism of these flavonoids. Another strategy of antimicrobial activity of kaempferol and quercetin is associated with bacterial membrane disruption. It is reported that flavonoids could inhibit bacterial growth in several ways such as by membrane rupture and permeabilization, interact with membrane protein directly, uncontrolled ROS generation and inhibition of membrane and cell wall biosynthesis [78]. The screening of kaempferol and quercetin activity against *E. coli* and *S. aureus* revealed cell rupture in both the bacteria [78]. In light of these literature findings, both kaempferol and quercetin can be taken into consideration as an essential phytochemical in the development of new antibiotic against various microbes.

## 6. Anti-Fungal Activity of Kaempferol and Quercetin

Kaempferol and quercetin antifungal activities are not well documented as compared to their antibacterial activity. Both flavonoids inhibit the growth of certain fungus such as, *Aspergillus fumigatus*, *Aspergillus niger*, *Candida. Albicans*, and *Saccharomyces cerevisiae* [79,80]. In some cases, they indirectly affect the fungal growth by sensitizing other drugs. It was reported that individually, quercetin showed no effect on fungal growth but in combination with amphotericin B, quercetin significantly induced the antifungal activity of amphotericin B in response to *C. neoformans* [81]. Studies have shown that kaempferol possesses stronger anti-fungal activity than quercetin. Recently, it has been investigated that kaempferol has a minimal inhibitory concentration (MIC) of 256 µg/mL while quercetin has lower than 128 µg/mL against *C. albicans*, *C. tropicalis*, and *Cryptococcus neoformans* [81,82,83]. Little research work has been conducted on the mechanisms of antifungal activity of both the flavonoids. However, a recently reported study evaluated that the anti-fungal mechanism of kaempferol and quercetin mainly includes disruption of the plasma membrane and affects the nucleic acid synthesis, protein synthesis and inhibits mitochondrial function [84]. Researchers have also studied the inhibition of fungus biofilm formation by both flavonoids. They observed that exposure of *C. orthopsilosis* and *C. metapsilosis* to both flavonoids reduces the metabolic activity and biomass of the two fungus [85]. These results validate recent studies that have demonstrated that kaempferol and quercetin inhibit microbial biofilm formation, possibly by weakening the cellular adhesion to abiotic surfaces, as reported for *S. aureus* and for *C. albicans* [72,86]. It has also been studied that mature biofilm of Candida specie is tolerant to flavonoids [83,86]. Kaempferol increased the metabolic activity of *C. parapsilosis sensu stricto* and *C. orthopsilosis*, while quercetin decreased it only for *C. parapsilosis sensu stricto* [85]. These differences in metabolic response do not necessarily reflect an alteration in the number of viable cells, since the metabolic activity of biofilms vary throughout the stages of biofilm formation [87], and may be a response to the stress caused by flavonoids exposure.

## 7. Bioactivity of Kaempferol and Quercetin in Human Cardiovascular Disease

The literature review showed that limited data are present regarding clinical trials on the bioactivity of kaempferol and quercetin. However, epidemiological studies regarding kaempferol have investigated in association with cardiovascular health issue (no clinical proof) [18]. The bioactivity of kaempferol in humans depends on the ingested conjugate however, the bioavailability of dietary kaempferol in human has not yet been reported. Oxidation of low-density lipoprotein (LDL cholesterol) is atherogenic and consider important in the formation of atherosclerotic plaques. Oxidized LDL form foam cells in the arteries which leads to the formation of atherosclerotic plaques [88]. Oxidized LDL induce cells of the arteries wall to form chemotactic factors, adhesion molecules, growth factors and cytokines, which enhance the development of plaque [89,90]. Kaempferol and quercetin act as inhibitors of oxidation of low-density lipoproteins via scavenge ROS which prevent or slow down this chain of actions and may reduce the risk of coronary heart complications. Another study investigated that platelets aggregation contributes to atherosclerotic disease development and acute platelets thrombus formation while quercetin act as inhibitor of platelets aggregation [91]. Kaempferol and quercetin are predicted to be supportive of mitochondrial function in cardiac cells. Ischemia-reperfusion (low blood supply) results into tissue damage depending on the duration of ischemia-reperfusion. Researchers reported that mitochondrial damage induced by ischemia-reperfusion is cause by ROS generation [92,93]. The indirect evidence show that the free radical scavengers (Such as kaempferol, quercetin) protect the cardiovascular from the destructive effects of ROS [93,94,95].

The first study on dietary intake of flavonoid related to mortality from coronary heart disease was studied in human in 1993 [96]. According to this report, the average daily intake of flavonoid was about 25.9 mg which was inversely associated with mortality from coronary heart disease [96]. The previously reported evidence shows that the antioxidant effect of quercetin affords protection of brain, heart, and other tissues in response to ischemia-reperfusion injury and other factors that induce oxidative stress [97]. Changes in the red blood cells deformability has a key role in various disease especially in cardiovascular disease [98]. The application of quercetin to the diabetic rats maintained the red blood cells deformability which is in agreement with the previous study showing the increase in red blood cells membrane fluidity after quercetin treatment in the hypercholesterolemic patient [99,100]. Quercetin protect erythrocyte cells by scavenging the ROS and reduces the formation of acanthocytes during oxidative stress [101]. Another study associated with flavonoid and myocardial infarction and fatal coronary heart disease investigated that, kaempferol from broccoli and tea was inversely associated with coronary heart disease with a relative risk of 0.66 (95% CI: 0.48–0.93, *p* = 0.04), while non association was found with myocardial infarction [102]. Another study concluded that, the higher dietary intake of kaempferol significantly reduced the inflammation in human [103]. At the same time, there is no evidence of toxicity of oral intake of kaempferol in human. However, it is reported that overdose of kaempferol may cause self-oxidation in human but has no effects on animals [104,105].

The clinical trial of quercetin bioactivity is also limited. However, a report shows that quercetin (1 g) increases the plasma from 0.1 to1.5 µmol/L without any cardiovascular or thrombogenic risk [106]. Higher amount (730 mg) of quercetin aglycon can reduce high blood pressure in the initial stage, but at the later stage of blood pressure, lower amount (150 mg) of quercetin aglycon can reduce blood pressure [107,108]. Another report shows that 500 gm significantly lower the inflammatory markers like TNF-α (tumor necrosis factor-α) and IL-6 (interleukin-6) [18]. Further studies are needed to investigate the bioactivity of quercetin metabolites on inflammatory markers in patients with elevated markers and at high risk of cardiovascular diseases. In human studies, oral uptake of quercetin was administered as a purified aglycon and the recommended dose is range from 150–5000 mg/day for maximum 12 weeks [18]. Literature review shows that, quercetin metabolism take place in the kidney, therefore two human studies associated to quercetin has been identified so far, relating to liver or kidney biomarker safety. It was investigated that, 150 mg/day intake of quercetin aglycon has no significant change in liver, kidney, hematology, or electrolytes biomarkers which indicating a daily dose of 150 mg was safe [108]. The high amount (5000 mg/day) of quercetin supplemented for 4 weeks did not cause adverse events. In 2010, quercetin supplements were added to the Food and Drug Administration’s Generally Recognized as Safe (GRAS) list for use as a supplemental ingredient added in foods and beverages up to 500 mg per serving [18].

Although there are several benefits and no health issue in the use of kaempferol and quercetin, however, there are several limitations to significantly conclude that the dietary intake of kaempferol and quercetin reduces inflammation in human. As the intake of kaempferol and quercetin are mainly from fruits and vegetables which also contains other bioactive compounds. These compounds might be more effective than kaempferol and quercetins against mentioned diseases. Therefore, the bioactivity and metabolism of kaempferol and quercetin in the human body is needed to further investigate and to better understand the mechanism of action of both the flavonoids.

## 8. Pharmacokinetic Characteristics of Kaempferol and Quercetin

During the last few years, a great deal of data has been published that points to the health-promoting properties of plant derived flavonoids. However, kaempferol and quercetin are the two major representative of flavonol that have received more attention during the last few years. In order to better understand the pharmacological effects of both the kaempferol and quercetin, pharmacokinetic studies are required [109]. Most of the flavonoids absorbed from common foods are present in plasma in the form of metabolites conjugated with methyl, glucoronate and sulfate group. Reports show that, flavonoid conjugates are hydrolysed to aglycone by enterobacteria before passing through the small intestine, while flavonoid aglycone can be directly absorbed into epithelial cells in the intestine [110,111]. Analytically quantitative determination of kaempferol and quercetin in a biological sample are the main determinants of pharmacokinetic study. Among the kaempferol and quercetin, the mechanism of bioavailability of quercetin is well studied. Reports show that, very small amount of quercetin absorbed in the stomach while large amount is absorbed in the small intestine and pass through the epithelial cells and reach the circulatory system [112,113]. Quercetin conjugates are difficult to pass through cellular membrane due to lipid bilayer with high membrane polarity [114], and they need transporter to pass through the membrane. The common transporter that transport the quercetin and its glycosides through the membrane, are the sodium dependent glucose co-transporter present on the apical membrane of intestinal epithelial cells [115]. Organic anion transport polypeptides is another solute carrier protein present in the intestinal epithelial cells that facilitate the polar compound influx into the cell [116]. It is reported that quercetin is a substrate for organic anion transport polypeptides taken up into intestinal epithelial cells [116]. Further, it is found that deglycosylation of quercetin glycosides occur in the intestine increases intestinal absorption, which results into increases plasma concentrations and enhance the bioavailability [117]. Lactose phlorizin hydrolase is an enzyme found in mammalian small intestine which is related to quercetin glycosides deglycosylation. For example, quercetin-*4*′-glucoside and quercetin-*3*-glucoside are the substrate for the lactose phlorizin hydrolase which hydrolysed them into quercetin aglycone [118]. These aglycone of quercetin are lipophilic (fat soluble) in nature and easily absorbed by intestinal epithelial cells by passive diffusion. It is reported that, quercetin *4*′-*β*-glucoside and quercetin-*3*-glucoside can be integrally absorbed by intestinal epithelial cells mediated by sodium dependent glucose co-transporter and then hydrolysed into quercetin aglycone by cytosolic *β*-glucosidase [119,120,121]. Similarly, kaempferol is also a lipophilic compound like the other flavonoids and mostly absorbed in to body through small intestine. Due to its lipophilicity, it is absorbed by passive diffusion, facilitated diffusion and active transport [122]. Similar to quercetin, kaempferol also metabolized in the small intestine by the intestinal enzymes. Researchers investigated that the flora of intestinal colon metabolizes the glycosides of kaempferol into aglycones and convert them into 4-methylphenol, 4-hydroxyphenylacetic acid and phloroglucinol which are capable to absorbed into circulation system and distributed into different tissues [123,124].

To date, the analysis of kaempferol and quercetin has been determined by thin-layer chromatography (TLC), gas chromatography (GC), capillary electrophoresis (CE), high-performance liquid chromatography (HPLC) [125]. Several studies showed that, flavonoids are converted into inactive (glucuronides) form in the gut resulting into reduced concentration of active (aglycones) form [126,127,128]. However, other studies evaluated multiple biological actions of flavonoid glucuronides and have found that beta-glucoronidase-expressing macrophages convert inactive glucoronides into active aglycones at local lesions with inflammation [129,130,131]. Xin Jia et al. reported that metabolism of kaempferol, quercetin and some other flavonoids were faster in gastric ulcer rats than that in the normal rats [132]. They suggested that the faster metabolism and transport of these flavonoids in the gastric ulcer rats might be due to the flavonoids concentration and consumption in the lesion site. Another pharmacokinetic study was conducted on bioavailability of kaempferol and quercetin using *Ginkgo biloba* extract (GBE), *Gingko biloba* phospholipid complexes (GBP) and *Gingko biloba* extract solid dispersions (GBS) in rat. The results indicated that, the bioavailability of kaempferol and quercetin in rat was increased after oral dosing of GBP and GBS comparing with GBE [133]. In some cases, of pharmacokinetic characterization of kaempferol and quercetin, double peaks is produced which is suggested to be due to some factors such as, enterogastric circulation, enterohepatic circulation, or due to the drug distribution in vivo [134,135,136]. The pharmacokinetic behavior of individual kaempferol and quercetin are sometimes different from their complex form. Li et al., studied pharmacokinetic characteristics of *Hippopae rhamnoides* L. through liquid chromatography-mass spectrometric (UPLC-MS) using the rat plasma [137]. Their results suggested that the pharmacokinetic behaviors of isorhamnetin, kaempferol and quercetin when administered together in a complex herbal extract might be different than the individual behaviors of the same compounds administered in their pure forms. They further investigated that kaempferol and quercetin was absorbed by passive diffusion in rats and no double peak were detected by UPLC-MS analysis. Quercetin can quickly metabolize to its corresponding glucuronides after successful absorption however, kaempferol absorb more efficiently in human than the quercetin [17,138,139,140]. Pharmacokinetic comparison between quercetin and quercetin *3*-*O*-*β*-glucuronide in rats revealed that quercetin *3-O*-*β*-glucuronide showed more delayed absorption than quercetin aglycone, as the glucoside/glucuronide forms of quercetin are too polar to cross cellular membranes by diffusion, hampering their absorption and bioavailabilities [141]. These findings may be explained by the hydroxylation of *3-O-β*-glucuronide into quercetin aglycone by luminal lactase phlorizin hydrolase or by intestinal microflora before absorption into blood. Another study demonstrated that *3-O*-glucosylation of quercetin increases its absorption while, bonding of a rhamnose to the aglycone inhibits its absorption [142].

## 9. Conclusions and Future Perspective

Our literature review concluded that kaempferol and quercetin efficiently protect plants during stress condition via regulation of certain hormones and secondary metabolites. The pharmacokinetic study evaluated its importance in the human health. From the literature study, it is concluded that they reduce LDL oxidation and platelets aggregation, which results into inhibition of formation of atherosclerotic plaques. They also protect against mitochondrial damage in the cardiac cells, which reduces cardiovascular complications. Their pharmacokinetic study can improve the understanding of their bioavailability, bioactivity, and metabolism. Previously, their antibacterial, antifungal and anti-oxidative properties were mostly studied, and further evaluation in term of emerging as regulatory and signalling molecule is still needed. Both the flavonoids possess antifungal and bacterial properties, and therefore, it is needed to significantly study their role as antibiotics in human health.

## Figures and Tables

**Figure 1 plants-11-02623-f001:**
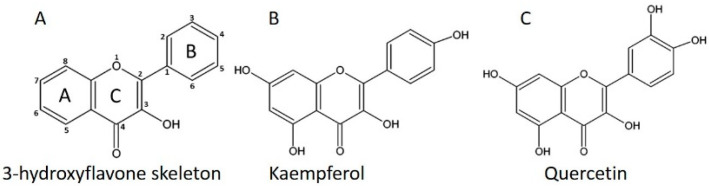
Chemical structure of 3-hydroxyflavone, kaempferol and quercetin. (**A**) shows flavonol skeleton, (**B**) shows kaempferol and (**C**) shows quercetin structure.

**Figure 2 plants-11-02623-f002:**
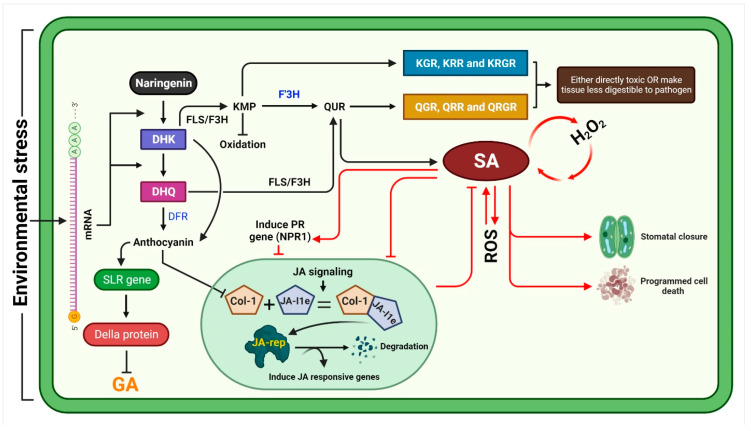
Proposed flow chart of the regulation of plant defence system mediated via kaempferol and quercetin under stress condition. The figure is adopted from our previously published article [50]. DHK (dihydrokaempferol), DHQ (dihydroquercetin), DFR (dihydroflavonol *4*-reductase), FLS (flavonol synthase), F3H (flavanone 3-hydroxylase), KMP (kaempferol), QUR (quercetin), QGR (quercetin *3-O*-glucoside *7-O*-rhamnoside), QRR (quercetin *3-O*-rhamnoside *7-O*-rhamnoside), QRGR (quercetin *3-O*-[*6*″-*O*-(rhamnosyl) glucoside] *7-O*-rhamnoside), SLR (slender rice mutant), GA (gibberellic acid), JA (jasmonic acid), Col-1 (collagen type I), JA-l1e (Jasmonic Acid-Isoleucine), PR (pathogenesis related), NPR (nonexpressor pathogenesis-related), KGR (kaempferol *3-O* glucoside *7-O* rhamnoside), KRR (kaempferol-3,7-dirhamnoside), KRGR (kaempferol *3-O-*[*6*″-*O*-(rhamnosyl) glucoside] *7-O* rhamnoside), SA (salicylic acid), and ROS (reactive oxygen specie).

**Table 1 plants-11-02623-t001:** Total flavonoids, kaempferol and quercetin in selected edible plants.

Scientific Name	Total Flavonoids mg/kg	Kaempferol mg/kg	Quercetin mg/kg
*Brassica oleracea*/broccoli	197.0	ND	60.0
*Brassica oleracea*/cauliflower	219.0	ND	219.0
*Brassica alboglabra*	14.5	ND	14.5
*Capsicum annum*	829.0	ND	799.5
*Allium fistulosum*	2720.5	832.0	1497.5
*Allium sativum*	957.0	ND	47.0
*Carica papaya*	1264.0	453.0	811.0
*Anacardium occidentale*	450.5	DN	262.5
*Hibiscus esculentus*	260.0	DN	205.5
*Cucurbita maxima*	371.0	371.0	ND
*Daucus carota*	232.5	140.0	55.0
*Raphanus sativus*	65.0	38.5	17.5
*Amaranthus gangeticus*	29.5	ND	29.5
*Sesbania grandifolia*	306.0	21.0	18.5
*Sauropus androgynus*	785.0	323.5	461.5
*Hydrocotyle asiatica*	444.0	20.5	423.5
*Phaeomeria speciosa*	307.0	286.0	21.0
*Mentha arvensis*	48.5	ND	48.5
*Camellia chinensis*	1491	ND	1070.0
*Pachyrrhizus erosus*	37.0	37.0	ND

ND: not detected, the table is adopted from Koo Hui Miean and Suhaila Mohamed (2001) [31].

## Data Availability

The data presented in this study are available on request from the corresponding author.

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
