# Peer review of "Bioactivity and Therapeutic Potential of Kaempferol and Quercetin: New Insights for Plant and Human Health"

_plants, 2022, doi:10.3390/plants11192623_

Round 1
Reviewer 1 Report (Previous Reviewer 1)
The authors followed the suggestions; therefore, the article is now in a much better shape. Before acceptance, I recommend a thorough revision of the manuscript and of the English form.
Author Response
Thank you for giving us the opportunity to submit a revised draft of my manuscript. We appreciate the time and effort that you and the reviewers have dedicated to providing your valuable feedback on our manuscript. We are grateful to the academic editor for his/her insightful comments on this paper. We have been able to incorporate changes to reflect most of the suggestions provided by the reviewers and the academic editor and we feel that the reviewer and editor suggestions improved our manuscript. All the changes made, were highlighted with track changes.
Such as, we critically revised “ pharmacokinetic characteristics of kaempferol and quercetin” and incorporated possible information, we also revised “sugar glucosides absorbed quickly than the other glucosides” and TLC.
We also revised the English thoroughly by native english speaker. Thank you.

Reviewer 2 Report (New Reviewer)
In order to demonstrate the bioactivity, therapeutic potential, and potential of kaempferol and quercetin as signaling molecules for hormonal regulation in stressful environments, the authors summarize the role of kaempferol and quercetin in plants and in human health in terms of human antimicrobial activity, bioavailability, and bioactivity. Overall, the article is comprehensive, but there are still some shortcomings which need to be addressed.
1. The article proposes as kaempferol and quercetin as the most studied flavonoids because of their antioxidant capacity and role in the defense system, whether the description could be developed specifically for the antioxidant potential and role in the defense system to complement the article?
2. The bioavailability of kaempferol and quercetin is not adequately depicted for plant and human health, that is, the specific mechanisms are not clearly explained.
3. The paper proposes that kaempferol and quercetin play their biological activities as strong antioxidants, whether it is possible to increase the legend display.
4. Part 7 of the article only lists relevant literature studies on the biological activities of kaempferol and quercetin, which are not convincing and should clarify how their biological activities play a role in human cardiovascular disease.
5. The legend description of Figure 2 in the text should be supplemented and complete, refer to Figure 3 in the article, and Figure 2 should be an original picture, not directly citing other literature pictures.
6. Abbreviations should be added below the legends in the text.
7. A list of abbreviated words may be added to the article to increase its completeness.
8. The conclusion and outlook section of the article should summarize the role and research significance of kaempferol and quercetin in terms of human antimicrobial activity, bioavailability and bioactivity for plant and human health, and further point out the future research directions to focus on.
Author Response
- The article proposes as kaempferol and quercetin as the most studied flavonoids because of their antioxidant capacity and role in the defense system, whether the description could be developed specifically for the antioxidant potential and role in the defense system to complement the article?
Reply: Thank you for your valuable comment. Actually, both the flavonoids has other essential roles such as fruit color, plant development, cell growth and development. However, here in the current review, we only focused on their antioxidant potential and defense system because being efficient antioxidants, kaempferol and quercetin has a key role in plant and human health. Therefore, we believe that focusing specifically on their antioxidant activity and role in defense system can be sufficient for the article.
- The bioavailability of kaempferol and quercetin is not adequately depicted for plant and human health, that is, the specific mechanisms are not clearly explained.
Reply: The information related to kaempferol and quercetin bioavailability in human was added to section 8, however the mechanism of bioavailability in plants is not well evaluated in the literature.
- The paper proposes that kaempferol and quercetin play their biological activities as strong antioxidants, whether it is possible to increase the legend display.
Reply: We are sorry, we didn’t understand what the reviewer mean by “legend display”.
- Part 7 of the article only lists relevant literature studies on the biological activities of kaempferol and quercetin, which are not convincing and should clarify how their biological activities play a role in human cardiovascular disease.
Reply: Thank you for your valuable evaluation, we added some details of their biological activities related to cardiovascular disease.
- The legend description of Figure 2 in the text should be supplemented and complete, refer to Figure 3 in the article, and Figure 2 should be an original picture, not directly citing other literature pictures.
Reply: Thank you for your valuable comment, the description of figure 2 were added into the text. However, there is a minor modification in the figure 2 and we published it previously so there is no copyright issue in the figure.
- Abbreviations should be added below the legends in the text.
Reply: Abbreviation added.
- A list of abbreviated words may be added to the article to increase its completeness.
Reply: Thank you for your suggestion, although we provided the list of abbreviated words at the end of the manuscript bellow the reference however, the acronyms of the abbreviated words are given where it is mentioned for the first time.
- The conclusion and outlook section of the article should summarize the role and research significance of kaempferol and quercetin in terms of human antimicrobial activity, bioavailability and bioactivity for plant and human health, and further point out the future research directions to focus on.
Reply: Thank you for your valuable comment, we revised the conclusion accordingly.
The changes suggested by reviewer 1 are highlighted in green color in the manuscript.
Reviewer 3 Report (New Reviewer)
The topic of review Bioactivity and therapeutic potential of kaempferol and quercetin: New insights for plant and human health proposed for publication on Plants fits well with the scopes of the Journal. The manuscript gives a wide sight on flavonoids kaempferol and quercetin, underscoring their biological potential. In particular, the authors addressed their attention to the effect of flavonoids in plants and, in addition, focused on their potential beneficial effect on human health.
From my side, my review process can mainly cover this latter feature of flavonoids, as plants and their physiology are not within my expertise area.
I only suggest minor revisions, such as:
The review needs a check of English language.
I suggest to more clearly divide the review into two parts, one dealing with plants health and the other one with human health, so to make the reader go more easily through the text.
I suggest to add the following reference, Priyanka Singh, Yamshi Arif, Andrzej Bajguz, Shamsul Hayat. 2021. The role of quercetin in plants DOI: 10.1016/j.plaphy.2021.05.023, to improve the Introduction.
With regard to quercetin, I would suggest to introduce a very recent study about the effect of quercetin on red blood cells, to enrich “cardiovascular section” with novel findings.
-Tomas Jasenovec, Dominika Radosinska, Marta Kollarova, Peter Balis, Kristina Ferenczyova, Barbora Kalocayova, Monika Bartekova, Lubomira Tothova, Jana Radosinska. 2021. Beneficial Effect of Quercetin on Erythrocyte Properties in Type 2 Diabetic Rats. DOI: 10.3390/molecules26164868
-A Remigante; S Spinelli; N Basile; D Caruso; G Falliti; S Dossena; A Marino; R Morabito. 2022. Oxidation Stress as a Mechanism of Aging in Human Erythrocytes: Protective Effect of Quercetin. Int. J. Mol. Sci. 2022, 23(14):7781. https://doi.org/10.3390/ijms23147781,
-Alessia Remigante, Sara Spinelli, Elisabetta Straface, Lucrezia Gambardella, Daniele Caruso, Giuseppe Falliti, Silvia Dossena, Angela Marino, Rossana Morabito. 2022. Antioxidant Activity of Quercetin in a H2O2-induced Oxidative Stress Model in Erythrocytes: Functional Role of Band 3 Protein. Int J Mol Sciences, section: Bioactives and Nutraceuticals, Dietary Antioxidants for Health and Longevity, 23, 10991. https://doi.org/10.3390/ ijms231910991
As a whole, I consider the review suitable for publication upon the above suggested minor changes.
Author Response
The review needs a check of English language.
Reply: thank you for your suggestion, the whole manuscript we check previously two times with native English speaker.
I suggest to more clearly divide the review into two parts, one dealing with plants health and the other one with human health, so to make the reader go more easily through the text.
Reply: thank you for your suggestion and we really appreciate your suggestion, however the manuscript dealing with plants, human, fungi and bacteria and at this stage we think that it is not convenient to divide the manuscript in different parts.
I suggest to add the following reference, Priyanka Singh, Yamshi Arif, Andrzej Bajguz, Shamsul Hayat. 2021. The role of quercetin in plants DOI: 10.1016/j.plaphy.2021.05.023, to improve the Introduction.
Reply: thank you for your suggestion, we added some more information in the introduction section from the suggested review article.
With regard to quercetin, I would suggest to introduce a very recent study about the effect of quercetin on red blood cells, to enrich “cardiovascular section” with novel findings.
-Tomas Jasenovec, Dominika Radosinska, Marta Kollarova, Peter Balis, Kristina Ferenczyova, Barbora Kalocayova, Monika Bartekova, Lubomira Tothova, Jana Radosinska. 2021. Beneficial Effect of Quercetin on Erythrocyte Properties in Type 2 Diabetic Rats. DOI: 10.3390/molecules26164868
-A Remigante; S Spinelli; N Basile; D Caruso; G Falliti; S Dossena; A Marino; R Morabito. 2022. Oxidation Stress as a Mechanism of Aging in Human Erythrocytes: Protective Effect of Quercetin. Int. J. Mol. Sci. 2022, 23(14):7781. https://doi.org/10.3390/ijms23147781,
-Alessia Remigante, Sara Spinelli, Elisabetta Straface, Lucrezia Gambardella, Daniele Caruso, Giuseppe Falliti, Silvia Dossena, Angela Marino, Rossana Morabito. 2022. Antioxidant Activity of Quercetin in a H2O2-induced Oxidative Stress Model in Erythrocytes: Functional Role of Band 3 Protein. Int J Mol Sciences, section: Bioactives and Nutraceuticals, Dietary Antioxidants for Health and Longevity, 23, 10991. https://doi.org/10.3390/ ijms231910991
Reply: thank you for sharing some informative articles, we added further informations in the section 7 from the suggested articles.
The changes suggested by reviewer 2 are highlighted in yellow color in the manuscript.
Round 2
Reviewer 2 Report (New Reviewer)
The points have been well-addressed.
This manuscript is a resubmission of an earlier submission. The following is a list of the peer review reports and author responses from that submission.
Round 1
Reviewer 1 Report
This manuscript explores the beneficial potential of two flavonoids, quercetin and kaempferol, contained in different quantities in plants. The mediated action of the two molecules by the accumulation of salycilic acid against pathogens is highlighted, in particular kaempferol acts by reducing the adhesion of fibrinogen and therefore the formation of biofilm. Both interact with bacterial DNA helicase by inhibiting its ATPase activity and thus bacterial growth. The two flavonoids also have fungicidal activity expressed through inhibition of protein synthesis and mitochondrial function. In human health quercetin and kaempferol seem to reduce mortality from coronary heart disease and have anti-inflammatory activities, although further human studies are needed the review highlights the versatile biological activities of the two flavonoids.
The review is interesting, but the discussion appears confusing, with several language errors and in some cases, it is challenging for the reader to grasp the message the authors want to convey.
So, I recommend a careful review of the manuscript and language editing before publication.
Some specific comments are as follows:
Page 1 in the Abstract line 20 to 24 should be reworded.
Page 2 line 64 to 71 the sentence is too long.
Similar sentences and meaning practically the same are written in lines 125 to 136.
Page 4 “General mechanism of stress inhibition by kaempferol and quercetin” paragraph should be redacted in a more careful way. For instance,
Line 141 Replace "has" for "have".
Line 144 to 152 the meaning of the sentences is lost. Please, rephrase.
Line 144 the definition of acronym "ROS" is already present on page 2.
In general, the definitions of many acronyms such as Kr, Qu, JA, GA, SLR and so on are missing in the text.
Page 6 line 220 to 222 the meaning of the sentence is lost. Please, rephrase.
Line 222 replace "act" for “acts”.
Line 226 replace "genes expression" for “gene expression”.
Line 232 replace "inhibit" for “inhibits”.
Line 236 to 239 the meaning of the sentence is lost. Please, rephrase.
Line 242 replace "literatures review" for “literature reviews”.
Line 249 replace "inhabit" for “inhibit”.
Line 256 replace "possess" for “possesses”.
Line 257 please, explains the meaning of “MICs value range”
Line 262 to 265, all verbs “include”, “affect”, “inhibit” should be in the singular.
Line 289 replace "flavonoid” for “flavonoids”.
Page 7 line 308 to 320 the meaning of the sentences is lost. Please, rephrase.
Page 8 line 329 to 337 the meaning of the sentences is lost. Please, rephrase.
Page 8 many definitions of acronyms should have been included in the preceding paragraphs.
Author Response
This manuscript explores the beneficial potential of two flavonoids, quercetin and kaempferol, contained in different quantities in plants. The mediated action of the two molecules by the accumulation of salicylic acid against pathogens is highlighted, in particular kaempferol acts by reducing the adhesion of fibrinogen and therefore the formation of biofilm. Both interact with bacterial DNA helicase by inhibiting its ATPase activity and thus bacterial growth. The two flavonoids also have fungicidal activity expressed through inhibition of protein synthesis and mitochondrial function. In human health quercetin and kaempferol seem to reduce mortality from coronary heart disease and have anti-inflammatory activities, although further human studies are needed the review highlights the versatile biological activities of the two flavonoids.
The review is interesting, but the discussion appears confusing, with several language errors and in some cases, it is challenging for the reader to grasp the message the authors want to convey.
So, I recommend a careful review of the manuscript and language editing before publication.
Some specific comments are as follows:
Page 1 in the Abstract line 20 to 24 should be reworded.
Ans: Thank you, we revised the suggested lines and now it is easily understandable to readers.
Page 2 line 64 to 71 the sentence is too long.
Ans: Thank you for your kind suggestion, we revised the sentence accordingly.
Similar sentences and meaning practically the same are written in lines 125 to 136.
Ans: Thank you for your comment, we revised the suggested sentences.
Page 4 “General mechanism of stress inhibition by kaempferol and quercetin” paragraph should be redacted in a more careful way. For instance,
Line 141 Replace "has" for "have".
Ans: Has been done, thank you.
Line 144 to 152 the meaning of the sentences is lost. Please, rephrase.
Ans: Revised the sentence now it is clear, thank you.
Line 144 the definition of acronym "ROS" is already present on page 2.
Ans: Has been revised.
In general, the definitions of many acronyms such as Kr, Qu, JA, GA, SLR and so on are missing in the text.
Ans: We define all the acronyms throughout the manuscript where it mentioned at first time.
Page 6 line 220 to 222 the meaning of the sentence is lost. Please, rephrase.
Ans: Rephrased the sentence, thank you.
Line 222 replace "act" for “acts”.
Ans: has been done.
Line 226 replace "genes expression" for “gene expression”.
Ans: has been done.
Line 232 replace "inhibit" for “inhibits”.
Ans: has been done.
Line 236 to 239 the meaning of the sentence is lost. Please, rephrase.
Ans: Sentence has been rephrased, thank you.
Line 242 replace "literatures review" for “literature reviews”.
Ans: has been done.
Line 249 replace "inhabit" for “inhibit”.
Ans: has been done, thank you.
Line 256 replace "possess" for “possesses”.
Ans: has been done
Line 257 please, explains the meaning of “MICs value range”
Ans: Explained, thank you.
Line 262 to 265, all verbs “include”, “affect”, “inhibit” should be in the singular.
Ans: Has been done, thank you.
Line 289 replace "flavonoid” for “flavonoids”.
Ans: has been done.
Page 7 line 308 to 320 the meaning of the sentences is lost. Please, rephrase.
Ans: Thank you, we revised.
Page 8 line 329 to 337 the meaning of the sentences is lost. Please, rephrase.
Ans: revised, now it is easily understandable to reader.
Page 8 many definitions of acronyms should have been included in the preceding paragraphs.
Ans: thank you for your kind comments, we revised the manuscript carefully and resolved the acronyms issue.
Reviewer 2 Report
The manuscript “Antioxidant and therapeutic potential of kaempferol and quercetin: New insights for plant and human health” is a review which has the aim to summarize significant recent advances in terms of antioxidant and therapeutic potential of kaempferol and quercetin. A special attention is also devoted to their role in plant growth and defence regulation. A review specifically centered on recent findings in kaempferol, and quercetin could be interesting for the readers since could provide a useful close up about biological activities of these two flavonols, and it is new in the most recent literature. However, in my opinion, the manuscript would benefit from a major reorganization that I will further detail in the Major comment section. In brief, in the abstract authors specifically state that the review will focus on “bioavailability and bioactivity in human body as well as regulation of plant defence in response to stresses and signalling molecule in terms of hormonal modulation under stress conditions”. Nonetheless, only the section relative to plant stress regulation and defence seems to be appropriately structured and discussed. Bioavailability and bioactivity in human body are only partially reviewed in a single paragraph mainly focusing on anti-inflammatory properties of kaempferol and quercetin. Bioavailability of flavonoids is not thoroughly discussed, with all issues relative to absorption in the human gut and the glycosylation-dependence of the process. Finally, the authors choose to go into more detail about the antimicrobial activity, only partially describing the antioxidant potential. I think it is a good choice since some recent and extensive reviews were focused on the molecular mechanisms of flavonoids in vivo and in vitro antioxidant activity, detailing on all flavonoid classes [i.e.: N. Shen, T. Wang, Q. Gan, S. Liu, L. Wang, B. Jin, Plant flavonoids: Classification, distribution, biosynthesis, and antioxidant activity, Food Chemistry, Volume 383, 2022, 132531, ISSN 0308-8146, DOI: https://doi.org/10.1016/j.foodchem.2022.132531]. Therefore, if the authors choose to stay with this choice, I suggest modifying the title leaving out the term Antioxidant. The literature background is largely recent and adequate but should be implemented in some sections. The English language should be revised all through the text since several odd phrases and expressions occur.
Major comments
- Title: I suggest changing the review title leaving out the antioxidant potential
- Abstract: The authors should state clearly on which biological activity of flavonoids they would focus. The last sentence should be rephrased since it is not clear to the reader if the antioxidant and antimicrobial properties should be a part of the review work.
- Introduction: I would suggest the authors to add a more detailed description of flavonoids three-rings structure and of different subclasses. Seven subclasses, instead of six, should be listed (flavonols, anthocyanidin, flavanones, flavonols, isoflavones, flavones, chalcones) A detail on flavonols 3-hydroxyflavone backbone could be added. A figure with chemical structures of flavonols, kaempferol and quercetin could help in this case. In addition, in the Introduction only antioxidant activity of flavonoids is discussed and there are no mentions to bioavailability or other bioactivities in humans which are further discussed.
- Kaempferol and Quercetin in edible plants: In this paragraph the authors discuss the natural sources of kaempferol and quercetin. In my opinion more informations should be added here about the flavonoid differently glycosylated isoforms produced in plants. For a more comprehensive discussion, part of this paragraph should be devoted to the significant relationship between flavonoids glycosylation and absorption in the human gut, if the authors decide that bioavailability is one of the review focuses. Glycosylation, depending on its composition, can either promote or impair flavonoids absorption. In this light, also some hint about novel flavonoids nanoformulations to overcome bioavailability problems would be interesting. Some useful articles include:
Slámová K, Kapešová J, Valentová K. "Sweet Flavonoids": Glycosidase-Catalyzed Modifications. Int J Mol Sci. 2018 Jul 21;19(7):2126. doi: 10.3390/ijms19072126. PMID: 30037103; PMCID: PMC6073497.
Marín L, Miguélez EM, Villar CJ, Lombó F. Bioavailability of dietary polyphenols and gut microbiota metabolism: antimicrobial properties. Biomed Res Int. 2015;2015:905215. doi: 10.1155/2015/905215. Epub 2015 Feb 23. PMID: 25802870; PMCID: PMC4352739.
Hostetler GL, Ralston RA, Schwartz SJ. Flavones: Food Sources, Bioavailability, Metabolism, and Bioactivity. Adv Nutr. 2017 May 15;8(3):423-435. doi: 10.3945/an.116.012948. PMID: 28507008; PMCID: PMC5421117.
Kaushal, Niharika; Singh, Minni ; Singh Sangwan, Rajender, Flavonoids: Food associations, therapeutic mechanisms, metabolism and nanoformulations, Food Research International, 157July 2022, Article number 111442. 10.1016/j.foodres.2022.111442.
Huang M, Su E, Zheng F, Tan C. Encapsulation of flavonoids in liposomal delivery systems: the case of quercetin, kaempferol and luteolin. Food Funct. 2017 Sep 20;8(9):3198-3208. doi: 10.1039/c7fo00508c. PMID: 28805832.
- General mechanism of stress inhibition by kaempferol and quercetin: Figure 1 here reported only show naringenin and it should be modified including kaempferol and quercetin.
- Anti-bacterial activity of Kaempferol and Quercetin: I would suggest the authors to add some insight about the significant activity of kaempferol and quercetin in causing cell membrane leakage in both Gram positive and negative bacteria. In fact, this is well known as one of the main antibacterial molecular mechanisms of flavonoids. A growing number of studies also report the efflux pump inhibition activity, that is a key aspect in the synergic action of flavonoids and antibiotics. Some useful articles include:
Donadio G, Mensitieri F, Santoro V, Parisi V, Bellone ML, De Tommasi N, Izzo V, Dal Piaz F. Interactions with Microbial Proteins Driving the Antibacterial Activity of Flavonoids. Pharmaceutics. 2021 May 5;13(5):660. doi: 10.3390/pharmaceutics13050660. PMID: 34062983; PMCID: PMC8147964.
Ting Wu, Mengying He, Xixi Zang, Ying Zhou, Tianfu Qiu, Siyi Pan, Xiaoyun Xu, A structure–activity relationship study of flavonoids as inhibitors of E. coli by membrane interaction effect, Biochimica et Biophysica Acta (BBA) - Biomembranes, 1828,11, 2013, 2751-2756, ISSN 0005-2736, https://doi.org/10.1016/j.bbamem.2013.07.029.
Nguyen, T.L.A.; Bhattacharya, D. Antimicrobial Activity of Quercetin: An Approach to Its Mechanistic Principle. Molecules 2022, 27, 2494. https://doi.org/10.3390/molecules27082494.
- Anti-fungal of Kaempferol and Quercetin: the antimicrobial activity described against Staphylococcus aureus should be moved to the previous paragraph.
- Bioactivity of kaempferol and quercetin in Humans: The authors here mainly describe the therapeutic potential of kaempferol and quercetin in cardiovascular diasease. Probably it would be better to clearly state it in the paragraph title.
- Kaempferol and quercetin as growth regulators: This paragraph is devoted to flavonoids role as growth regulators in plants. Maybe it would be beneficial for the reader to move it after the paragraph “general mechanism of stress inhibition by kaempferol and quercetin”. In this way all paragraphs discussing flavonoids physiological roles in plants are in the first section.
- Conclusion and future perspective: Here some conclusions are drawn about flavonoids glucosides and derivatives activity and absorption. However, as previously stated, these features are not properly discussed in the text.
Minor comments
- Line 19: “defence” should be used in place “defense”, here and in all the manuscript.
- Line 22: there is a typo for “well”
- Line 24: Please rephrase “are still needed to further evaluate”
- Line 27: “signalling” should be used in place “signaling”, here and in all the manuscript
- Lines 34, 82, 139, 209, 246, 280, 338, 397: Paragraph numbers are missing.
- Line 53: It is not clear what authors means by “class of flavonols which differ”, please clarify
- Line 73: there is a typo for “recent”
- Line 120: Please cut the “are” in the sentence
- Line 148: there is a typo for “abiotic”
- Line 153: The authors used the abbreviations “Kr” and “Qu”. Please, add the extenso and define them the first time that appear in the text.
- Line 165: there is a typo for “generated”
- Line 220: The expression “study related microbes” is not clear, please rephrase.
- Lines 222-224: It should be stated that these evidences are relative to S. aureus.
- Line 249: there is a typo for “inhibit”
- Lines 256: “stronger” should be used in place of “strong”
- Line 316: there is a typo for “aglycon”
- Lines 329 – 334: Here the authors state that the cited studies were inconclusive in defining kaempferol and quercetin beneficial effects because their intake was mainly obtained by fruit and vegetables. I am not sure to understand the point because many studies are present in literature where purified flavonoids are administered. Maybe this should be better explained or reconsidered.
- Line 365: Please correct with “ABA is another phytormone reflecting”
- Line 397: there is a typo for “perspective”
- Line 404: “have” should be used in place of “has”
- Line 411: Please rephrase the expression “is still needed to be investigated”
Author Response
The manuscript “Antioxidant and therapeutic potential of kaempferol and quercetin: New insights for plant and human health” is a review which has the aim to summarize significant recent advances in terms of antioxidant and therapeutic potential of kaempferol and quercetin. A special attention is also devoted to their role in plant growth and defence regulation. A review specifically centered on recent findings in kaempferol, and quercetin could be interesting for the readers since could provide a useful close up about biological activities of these two flavonols, and it is new in the most recent literature. However, in my opinion, the manuscript would benefit from a major reorganization that I will further detail in the Major comment section. In brief, in the abstract authors specifically state that the review will focus on “bioavailability and bioactivity in human body as well as regulation of plant defence in response to stresses and signalling molecule in terms of hormonal modulation under stress conditions”. Nonetheless, only the section relative to plant stress regulation and defence seems to be appropriately structured and discussed. Bioavailability and bioactivity in human body are only partially reviewed in a single paragraph mainly focusing on anti-inflammatory properties of kaempferol and quercetin. Bioavailability of flavonoids is not thoroughly discussed, with all issues relative to absorption in the human gut and the glycosylation-dependence of the process. Finally, the authors choose to go into more detail about the antimicrobial activity, only partially describing the antioxidant potential. I think it is a good choice since some recent and extensive reviews were focused on the molecular mechanisms of flavonoids in vivo and in vitro antioxidant activity, detailing on all flavonoid classes [i.e.: N. Shen, T. Wang, Q. Gan, S. Liu, L. Wang, B. Jin, Plant flavonoids: Classification, distribution, biosynthesis, and antioxidant activity, Food Chemistry, Volume 383, 2022, 132531, ISSN 0308-8146, DOI: https://doi.org/10.1016/j.foodchem.2022.132531]. Therefore, if the authors choose to stay with this choice, I suggest modifying the title leaving out the term Antioxidant. The literature background is largely recent and adequate but should be implemented in some sections. The English language should be revised all through the text since several odd phrases and expressions occur.
Ans: Thank you so much for your effort and time dedicated to providing your valuable feedback on our manuscript.
Major comments
- Title: I suggest changing the review title leaving out the antioxidant potential
Ans: Thank you for your kind suggestion; we changed the title a little bit. Our main focus of the study is related to bioactivity and pharmaceutics of kaempferol and quercetin therefore complete changing of tittle will deviate study from the main aim.
- Abstract: The authors should state clearly on which biological activity of flavonoids they would focus. The last sentence should be rephrased since it is not clear to the reader if the antioxidant and antimicrobial properties should be a part of the review work.
Ans: thank you for your comment; actually, we mentioned that this study is focused on antimicrobial activity, therapeutic potential and their bioactivity in human body. We also rephrased the last sentence, I hope now it will be more clear.
- Introduction: I would suggest the authors to add a more detailed description of flavonoids three-rings structure and of different subclasses. Seven subclasses, instead of six, should be listed (flavonols, anthocyanidin, flavanones, flavonols, isoflavones, flavones, chalcones) A detail on flavonols 3-hydroxyflavone backbone could be added. A figure with chemical structures of flavonols, kaempferol and quercetin could help in this case. In addition, in the Introduction only antioxidant activity of flavonoids is discussed and there are no mentions to bioavailability or other bioactivities in humans which are further discussed.
Ans: thank you for your kind comment, detail classification of flavonoids are mention in our previous review paper, however we added some more detail about structure, bioavailability and bioactivity in human. We incorporated new figure (Figure 1) to the manuscript, illustrating the chemical structures of kaempferolkeampferol and quercetin and quercetin.
- Kaempferol and Quercetin in edible plants: In this paragraph the authors discuss the natural sources of kaempferol and quercetin. In my opinion more informations should be added here about the flavonoid differently glycosylated isoforms produced in plants. For a more comprehensive discussion, part of this paragraph should be devoted to the significant relationship between flavonoids glycosylation and absorption in the human gut, if the authors decide that bioavailability is one of the review focuses. Glycosylation, depending on its composition, can either promote or impair flavonoids absorption. In this light, also some hint about novel flavonoids nanoformulations to overcome bioavailability problems would be interesting. Some useful articles include:
Slámová K, Kapešová J, Valentová K. "Sweet Flavonoids": Glycosidase-Catalyzed Modifications. Int J Mol Sci. 2018 Jul 21;19(7):2126. doi: 10.3390/ijms19072126. PMID: 30037103; PMCID: PMC6073497.
Marín L, Miguélez EM, Villar CJ, Lombó F. Bioavailability of dietary polyphenols and gut microbiota metabolism: antimicrobial properties. Biomed Res Int. 2015;2015:905215. doi: 10.1155/2015/905215. Epub 2015 Feb 23. PMID: 25802870; PMCID: PMC4352739.
Hostetler GL, Ralston RA, Schwartz SJ. Flavones: Food Sources, Bioavailability, Metabolism, and Bioactivity. Adv Nutr. 2017 May 15;8(3):423-435. doi: 10.3945/an.116.012948. PMID: 28507008; PMCID: PMC5421117.
Kaushal, Niharika; Singh, Minni ; Singh Sangwan, Rajender, Flavonoids: Food associations, therapeutic mechanisms, metabolism and nanoformulations, Food Research International, 157July 2022, Article number 111442. 10.1016/j.foodres.2022.111442.
Huang M, Su E, Zheng F, Tan C. Encapsulation of flavonoids in liposomal delivery systems: the case of quercetin, kaempferol and luteolin. Food Funct. 2017 Sep 20;8(9):3198-3208. doi: 10.1039/c7fo00508c. PMID: 28805832.
Ans: thank you for your valuable suggestion. We incorporated the possible information.
- General mechanism of stress inhibition by kaempferol and quercetin: Figure 1 here reported only show naringenin and it should be modified including kaempferol and quercetin.
Ans: thank you for your valuable comment, however, we mentioned naringenin because it is a precursor of kaempferol and quercetin and we draw the mechanism started from transcriptional regulation.
- Anti-bacterial activity of Kaempferol and Quercetin: I would suggest the authors to add some insight about the significant activity of kaempferol and quercetin in causing cell membrane leakage in both Gram positive and negative bacteria. In fact, this is well known as one of the main antibacterial molecular mechanisms of flavonoids. A growing number of studies also report the efflux pump inhibition activity, that is a key aspect in the synergic action of flavonoids and antibiotics. Some useful articles include:
Donadio G, Mensitieri F, Santoro V, Parisi V, Bellone ML, De Tommasi N, Izzo V, Dal Piaz F. Interactions with Microbial Proteins Driving the Antibacterial Activity of Flavonoids. Pharmaceutics. 2021 May 5;13(5):660. doi: 10.3390/pharmaceutics13050660. PMID: 34062983; PMCID: PMC8147964.
Ting Wu, Mengying He, Xixi Zang, Ying Zhou, Tianfu Qiu, Siyi Pan, Xiaoyun Xu, A structure–activity relationship study of flavonoids as inhibitors of E. coli by membrane interaction effect, Biochimica et Biophysica Acta (BBA) - Biomembranes, 1828,11, 2013, 2751-2756, ISSN 0005-2736, https://doi.org/10.1016/j.bbamem.2013.07.029.
Nguyen, T.L.A.; Bhattacharya, D. Antimicrobial Activity of Quercetin: An Approach to Its Mechanistic Principle. Molecules 2022, 27, 2494. https://doi.org/10.3390/molecules27082494.
Ans: thank you for your kind suggestion. We incorporated further possible information in the suggested section.
- Anti-fungal of Kaempferol and Quercetin: the antimicrobial activity described against Staphylococcus aureus should be moved to the previous paragraph.
Ans: we deleted the Staphylococcus aureus from the “Anti-fungal of Kaempferol and Quercetin” section because its activity is already mentioned in the previous paragraph. Thank you.
- Bioactivity of kaempferol and quercetin in Humans: The authors here mainly describe the therapeutic potential of kaempferol and quercetin in cardiovascular diasease. Probably it would be better to clearly state it in the paragraph title.
Ans: revised the paragraph title, thank you.
- Kaempferol and quercetin as growth regulators: This paragraph is devoted to flavonoids role as growth regulators in plants. Maybe it would be beneficial for the reader to move it after the paragraph “general mechanism of stress inhibition by kaempferol and quercetin”. In this way all paragraphs discussing flavonoids physiological roles in plants are in the first section.
Ans: thank you for your valuable suggestion, we revised the paragraph title and rearranged the paragraph accordingly.
- Conclusion and future perspective: Here some conclusions are drawn about flavonoids glucosides and derivatives activity and absorption. However, as previously stated, these features are not properly discussed in the text.
Ans: thank you for your kind evaluation. According to your suggestion, we added more information about the glycosides, there bioactivity ion human body. We hope it will be now more clear to the readers.
Minor comments
- Line 19: “defence” should be used in place “defense”, here and in all the manuscript.
Ans: Replaced throughout the manuscript, thank you.
- Line 22: there is a typo for “well”
Ans: Revised, thank you.
- Line 24: Please rephrase “are still needed to further evaluate”
Ans: Rephrased, thank you.
- Line 27: “signalling” should be used in place “signaling”, here and in all the manuscript
Ans: Replaced throughout the manuscript.
- Lines 34, 82, 139, 209, 246, 280, 338, 397: Paragraph numbers are missing.
Ans: thank you, paragraph numbers were given.
- Line 53: It is not clear what authors means by “class of flavonols which differ”, please clarify
Ans: We revised the sentence, now it is more clear, thank you.
- Line 73: there is a typo for “recent”
Ans: revised, thank you.
- Line 120: Please cut the “are” in the sentence
Ans: revised accordingly, thank you.
- Line 148: there is a typo for “abiotic”
Ans: revised thank you.
- Line 153: The authors used the abbreviations “Kr” and “Qu”. Please, add the extenso and define them the first time that appear in the text.
Ans: thank you for your comment, we defined all the acronym where they mentioned first time.
- Line 165: there is a typo for “generated”
Ans: revised, thank you.
- Line 220: The expression “study related microbes” is not clear, please rephrase.
Ans: rephrased, thank you.
- Lines 222-224: It should be stated that these evidences are relative to S. aureus.
Ans: thank you for your comment, we included the name of S. aureus.
- Line 249: there is a typo for “inhibit”
Ans: revised.
- Lines 256: “stronger” should be used in place of “strong”
Ans: revised thank you.
- Line 316: there is a typo for “aglycon”
Ans: revised throughout the manuscript, thank you.
- Lines 329 – 334: Here the authors state that the cited studies were inconclusive in defining kaempferol and quercetin beneficial effects because their intake was mainly obtained by fruit and vegetables. I am not sure to understand the point because many studies are present in literature where purified flavonoids are administered. Maybe this should be better explained or reconsidered.
Ans: thank you for your comment. Actually the sentence was a bit confusing, we revised and now its more clear to the readers.
- Line 365: Please correct with “ABA is another phytormone reflecting”
Ans: correction has been done.
- Line 397: there is a typo for “perspective”
Ans: revised, thank you.
- Line 404: “have” should be used in place of “has”
Ans: revised, thank you.
- Line 411: Please rephrase the expression “is still needed to be investigated”
Ans: revised the sentence, thank you.
Reviewer 3 Report
This review was focused on kampferol and quercetin as common examples of flavonoids occur in plants. But all the activities reported in the literature are exerted by other related flavonoids. Therefore, authors should compare between the activity of kampferol and quercetin and that of other related flavonoids. The authors concluded that "In light of literatures review, both the kaempferol and quercetin can be taken into consideration as an essential phytochemical in the development of new antibiotic against various microbes." I think this conclusion required more evidents.
Author Response
This review was focused on kampferol and quercetin as common examples of flavonoids occur in plants. But all the activities reported in the literature are exerted by other related flavonoids. Therefore, authors should compare between the activity of kampferol and quercetin and that of other related flavonoids. The authors concluded that "In light of literatures review, both the kaempferol and quercetin can be taken into consideration as an essential phytochemical in the development of new antibiotic against various microbes." I think this conclusion required more evidents.
Ans: Thank you for your time dedicated to review our manuscript and we appreciate your valuable comments. You are right our review focused on kaempferol and quercetin, however in some cases we mentioned only general flavonoids because kaempferol and quercetin are not yet fully explored. Also, most of the mechanism of different types of flavonoids are common in all kind of flavonoids. However, we think that conclusion summed up all the informations discussed in the manuscript.
Round 2
Reviewer 1 Report
The authors have revised and improved the manuscript which can now be accepted.
However, it is recommended to specify the meaning of the acronyms that continue to be lost in the text despite the referee's warning: "In general, the definitions of many acronyms such as Kr, Qu, JA, GA, SLR and so on are missing in the text "
Author Response
Thank you so much for your valuable time to review our manuscript.
We carefully revised the whole manuscript and defined the acronyms throughout the manuscript.
The revision is highlighted in green color.
Reviewer 2 Report
In my opinion the manuscript is now acceaptable for pubblication in the present form after revisions. Only minor english corrections are required.
Author Response
Thank you for your kind suggestion.
We revised the manuscript by a native English speaker.
Reviewer 3 Report
The manuscript has fair contribution to the field and does not seem to be suitable for publication in plants.
Author Response
Thank you for the valuable time you dedicated to reviewing our manuscript.
We appreciate your comment, but we revised our manuscript extensively according to the other two reviewers' suggestions and comments and we hope now it is suitable for publication.